# Thalamic, cortical, and amygdala involvement in the processing of a natural sound cue of danger

**Ana G. Pereira** [ID]¤, **Matheus Farias, Marta A. Moita** [ID]*

Champalimaud Neuroscience Program at the Champalimaud Centre for the Unknown, Lisboa, Portugal

¤ Current address: Stanley Center for Psychiatric Research at Broad Institute, Cambridge, Massachusetts, United States of America
* marta.moita@neuro.fchampalimaud.org

**Data Availability Statement:** All relevant data will be within the paper and its Supporting Information files.

**Funding:** This work was developed with the financial support of the Champalimaud Foundation

## Abstract

Animals use auditory cues generated by defensive responses of others to detect impending danger. Here we identify a neural circuit in rats involved in the detection of one such auditory cue, the cessation of movement-evoked sound resulting from freezing. This circuit comprises the dorsal subnucleus of the medial geniculate body (MGD) and downstream areas, the ventral area of the auditory cortex (VA), and the lateral amygdala (LA). This study suggests a role for the auditory offset pathway in processing a natural sound cue of threat.

## Introduction

The use of auditory alarm cues from conspecifics is widespread across the animal kingdom [1–4]. Most research has focused on actively emitted signals, such as alarm calls and foot stamping [1,3]. However, auditory cues generated by movement patterns of prey can play a crucial role in predator–prey interactions. Although it is well known that predators use motion cues produced by moving prey for their detection [5–8], less is known about the ability of prey to use these motion cues to detect impending danger. Nevertheless, it has been established that crested pigeons use distinct wing whistles produced by conspecific escape flights [2] and rats use silence resulting from freezing as alarm cues [4]. Recently, it has also been suggested that seismic waves produced by fast running in elephants promote vigilance in conspecifics [9].

The neuronal pathways underlying defensive responses triggered by sounds have been extensively studied using classical conditioning to pure tones, sweeps, or white noise [10]. However, little is known about the mechanisms by which natural sounds (that most likely shaped the auditory system through evolution) trigger defensive responses [11,12]. Previously, we found that rats use freezing by others as a cue of danger. This is a learned cue, as it required prior experience with shock. Importantly, the cessation of the movement-evoked sound resulting from the onset of freezing was the alarm cue. In playback experiments, this cue was sufficient to trigger freezing in rats [4]. Hence, in the present study, we probed the neuronal circuits involved in the processing of natural sound cues of threat by testing the response of rats to the transition from movement-evoked sound to silence.

(https://m.fchampalimaud.org/en/champalimaud-research/); support to the Champalimaud Vivarium as part of the research infrastructure Congento, co-financed by Lisboa Regional Operational Programme (Lisboa2020), under the PORTUGAL 2020 Partnership Agreement, through the European Regional Development Fund (ERDF) and Fundação para a Ciência e Tecnologia (Portugal) under the project LISBOA-01-0145-FEDER-022170; MAM was supported by the European Research Council (ERC-2013-StG-337747 "C.o.C. O."); AP was supported by Fundação para a Ciência e Tecnologia (grant SFRH/BD/33943/2009). The funders had no role in study design, data collection and analysis, decision to publish, or preparation of the manuscript.

**Competing interests:** The authors have declared that no competing interests exist.

**Abbreviations:** ArchT, archaerhodopsin from *Halorubrum* strain TP009; Ct light, control light; Ct ArchT, control ArchT; CTB, cholera toxin B; DAPI, 4',6-diamidino-2-phenylindole; GFP, green fluorescent protein; LA, lateral amygdala; MGB, medial geniculate body; MGD, dorsal division of the MGB; MGV, ventral division of the MGB; MGM, medial division of the MGB; PBS, phosphate buffered saline; PD, posterodorsal area; PIL, posterior intralaminar thalamic nucleus; PP, peripeduncular nucleus; SG, suprageniculate nucleus; VA, ventral area.

## Results

First, we exposed individual rats to unsignaled footshocks, as prior studies found that rats previously exposed to shock but not naive ones respond to the distress of others [4,13]. The following day, we tested their freezing response to the cessation of movement-evoked sound (silence test) (Fig 1A). During the test, rats were placed in a box and allowed to explore for 3 minutes during which a speaker played the recorded sound of another rat moving [4] (see Methods). After this baseline period, the sound ceased for 1 minute (silence period). We started by looking at the role of the lateral nucleus of the amygdala (LA), broadly implicated in freezing driven by learned aversive sounds [10]. To this end, we optogenetically inactivated the LA using ArchT, a genetically encoded proton pump that hyperpolarizes neurons upon illumination with green light [14]. Importantly, the inactivation started 10 seconds before the onset of silence to encompass the transition from the movement-evoked sound to silence (Fig 1C and Methods). In this experiment, we used 3 groups: ArchT + light, in which neuronal activity is manipulated; and 2 control groups, control light (Ct light) and control ArchT (Ct ArchT), in which neuronal activity was left undisturbed (see Methods). Because there was no significant difference in freezing between the 2 control groups (S1 Fig), we combined them into a single

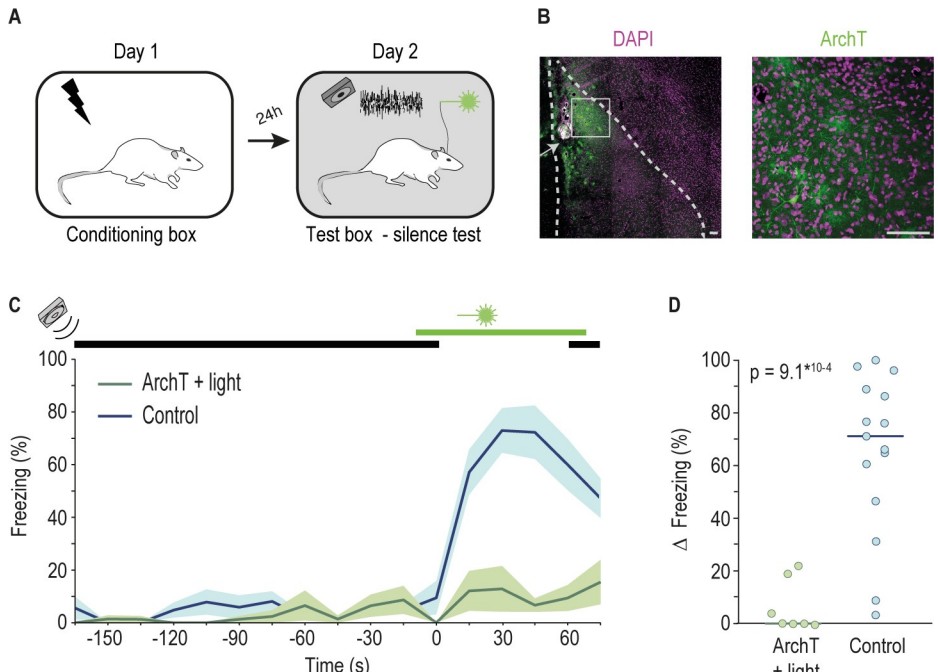

**Fig 1. Optogenetic inactivation of the LA abolishes freezing triggered by the cessation of the movement-evoked sound.** (A) Schematics of the behavioral paradigm. (B) Left, image of a coronal brain section from a representative rat, in which neurons in LA expressed ArchT; right, image depicting a higher magnification of the area highlighted by the white square on the left image. Pink, cell bodies labeled with DAPI. Green, cell bodies and cellular processes of cells expressing ArchT-GFP. White arrow indicates tip of the injector. Scale bar 100 μm. (C) Movement-evoked sound (indicated by black bar) was played to rats from the ArchT + light (green, *n* = 7) and Control (blue, *n* = 15 including animals from both Ct light and Ct ArchT) groups with an interval of 1 minute of silence (between 0 and 60 seconds). Green bar indicates the period during which 556-nm light was on. Line graph shows average % of freezing throughout the test session (shaded area shows dispersion of the data given by ±SEM) (S1 Data, Sheet Fig 1). (D) Individual dots represent the change in the percentage of time each animal spends freezing during the minute immediately preceding the cessation of the movement-evoked sound and the minute of silence. Horizontal bar represents the median value of the group (ArchT + light = 0.00%, Control = 71.20%, Wilcoxon rank sum test, rank sum = 33) (S1 Data, Sheet Fig 1). ArchT, archaerhodopsin from *Halorubrum* strain TP009; Ct ArchT, control ArchT; Ct light, control light; DAPI, 4', 6-diamidino-2-phenylindole; GFP, green fluorescent protein; LA, lateral amygdala.

control group. During the silence test, we found virtually no freezing during the baseline period in both ArchT + light and control animals. Silence onset led to a robust increase in freezing in the control group, whereas in the ArchT + light group, there were minimal changes in rats' freezing behavior (Fig 1C and 1D and S1 Fig). Similar results were found when analyzing the behavior of animals in the 2 control groups separately (Wilcoxon rank sum test comparing median change in freezing of ArchT + light and Ct light groups, $p = 0.003$, rank sum = 33; ArchT + light and Ct ArchT groups, $p = 0.001$, rank sum = 28) (also see S1 Fig). This result shows that activity in the LA is necessary for the display of freezing triggered by the transition from movement-evoked sound to silence.

The LA receives auditory information through a direct pathway originating in several subnuclei of the medial geniculate body (MGB) of the auditory thalamus [15]. These same MGB subnuclei are also the origin of an indirect pathway that involves connections through the auditory cortex as well as other cortical regions [10,15,16]. To study the auditory pathways that convey information about the cessation of movement-evoked sound to the LA, we started by asking which of MGB's subnuclei projecting to LA both directly and indirectly are active during exposure to the cessation of movement-evoked sound (Fig 2A). To this end, we quantified

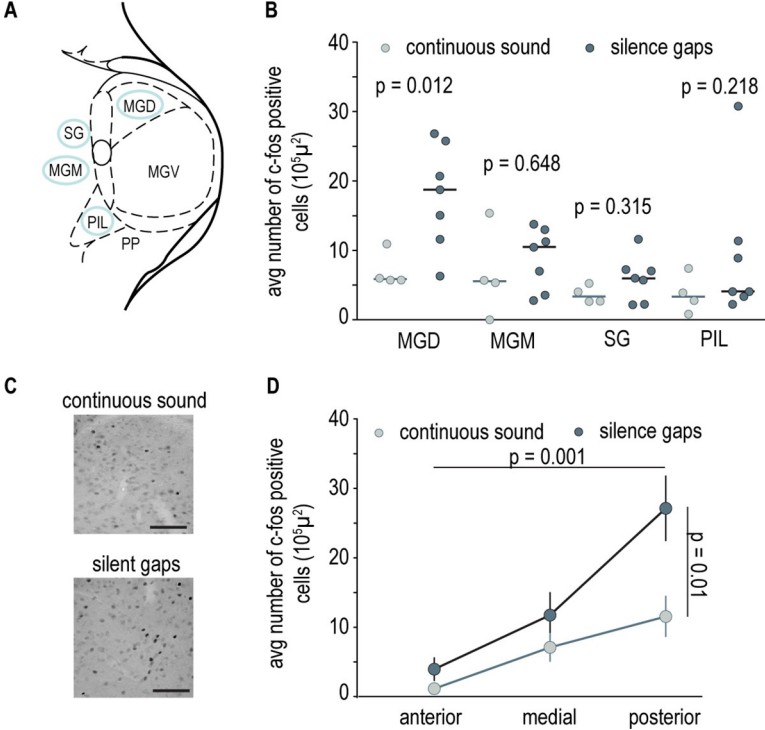

Fig 2. Differential activity in the MGD triggered by the cessation of movement-evoked sound. (A) Schematics of the MGB and its subnuclei. Regions circled in blue are part of both the mono- and polysynaptic pathway to the LA. (B) Average number of *c-fos*-positive cells in the MGD and MGM, the SG, and the PIL of individual rats from the continuous sound (light gray $n = 4$) and silence gaps (dark gray $n = 7$) group. The horizontal bar represents the median value, and Wilcoxon rank sum test was used for comparisons (MGD continuous sound = 5.87, silence gaps = 18.7, rank sum = 11; MGM continuous sound = 5.54, silence gaps = 10.50, rank sum = 21; SG continuous sound = 3.36, silence gaps = 5.95, rank sum = 18; PIL continuous sound = 3.33, silence gaps = 4.07, rank sum = 17) (S1 Data, Sheet Fig 2). (C) Representative pictures of c-*fos*-labeled cells in MGD. Scale bar 100 μm. (D) Line graph showing average number of *c-fos*-labeled cells in the MGD along the anterior-posterior axis for continuous sound and silence groups. Vertical bars represent ±SEM (*n*-way ANOVA main effect repeated measures AP axis $p = 0.001$, F = 12.69; silence versus continuous sound effect $p = 0.01$, F = 7.71) (S1 Data, Sheet Fig 2). LA, lateral amygdala; MGB, medial geniculate body; MGD, dorsal division of the MGB; MGM, medial division of the MGB; MGV, ventral division of the MGB; PIL, posterior intralaminar thalamic nucleus; PP, peripeduncular nucleus; SG, suprageniculate nucleus.

the expression of the neural activity marker *c-fos* in 2 groups of animals: one exposed to continuous playback of sound and another exposed to the same sound but with 2 periods of silence (see Methods). As expected, we found that rats exposed to the sound of movement with silence gaps increased freezing during the two 1-minute silence periods relative to the preceding minute (median change in freezing = 40.27%). No increase was observed when comparing the same time periods for rats in the continuous sound group (median change in freezing = −1.53%; Wilcoxon rank sum test comparing median change in freezing of silence gaps and continuous sound groups, *p* = 0.0303, rank sum = 53.5). The quantitative analysis of *c-fos* expression revealed that exposure to the sound of movement with silence gaps significantly increased the number of *c-fos*-positive cells in the dorsal division of the MGB (MGD) in comparison to continuous sound exposure (Fig 2B). In addition, we found that the number of *c-fos*-positive cells increased from anterior to posterior regions of the MGD (Fig 2D). We also observe that the average number of *c-fos*-labeled cells in the MGD of animals exposed to the sound of movement with silence gaps was quite variable. This variability is unlikely to result from differences in anatomic distribution of *c-fos*-positive cells within this subnucleus (such as proximity to other regions in the MGB that may also respond to sound offset), because we systematically probed the same region of interest within it (see Methods).

Our findings show that the MGD is the only subnucleus of the auditory thalamus projecting directly to LA that is more active when rats are exposed to the sound of movement with silence gaps. Prior electrophysiology studies in rodents report the presence of cells with sound offset responses throughout the MGB, with particular prevalence in the MGD (with the highest incidence in the caudal part) and the marginal zone, or shell, of the ventral division of the MGB (MGV) [17–20]. This suggests that *c-fos*-positive cells in the MGD may have been activated by the offset of the sound of movement when silence gaps are introduced. In agreement with this hypothesis, we also observed the presence of robust *c-fos* labeling in the ventral shell of the MGV, in particular, in more rostral areas (see S2 Fig). Alternatively, *c-fos* expression may reflect a response to an increase in stimulus saliency resulting from the introduction of silence gaps in the background sound. However, as we failed to see significant changes in the other MG subnuclei studied, this alternative explanation would imply that the MDG is the subnucleus with strongest sensitivity to changes in saliency. To our knowledge, there is no evidence in this direction. Future experiments in which the activity of MGD neurons is recorded during exposure to sound of movement with either silence gaps or other changes in saliency are necessary to definitively disambiguate between the 2 possibilities.

Next, we tested whether MGD activity is required for silence-triggered freezing by optogenetically inactivating this area (Fig 3A). We found that freezing was greatly reduced upon MGD inactivation relative to controls (Fig 3C and 3D). When analyzing the behavior of animals in the 2 control groups separately, the difference between ArchT + light and Ct ArchT was again significant; however, the difference relative to Ct light failed to reach significance (Wilcoxon rank sum test comparing median change in freezing of ArchT + light and Ct light groups, *p* = 0.07, rank sum = 40; ArchT + light and Ct ArchT groups, *p* = 0.017, rank sum = 31) (also see S3 Fig). However, if outliers are removed, then all comparisons reveal significant differences across groups (Wilcoxon rank sum test comparing median change in freezing of ArchT + light and Ct light groups, *p* = 0.038, rank sum = 27; ArchT + light and Ct ArchT groups, *p* = 0.0043, rank sum = 21). Several studies addressed the anatomical and physiological properties of the MGD [15,17–19,21–23]. However, to our knowledge, the contribution of this subnucleus to behavior remained untested. Therefore, we asked whether the MGD is required for freezing in response to a conditioned tone, as are other MGB subnuclei that project to LA [24]. To this end, we conditioned the same rats used in the previous experiment with 3 tone–shock pairings. The following day, we tested their response to the conditioned tone while

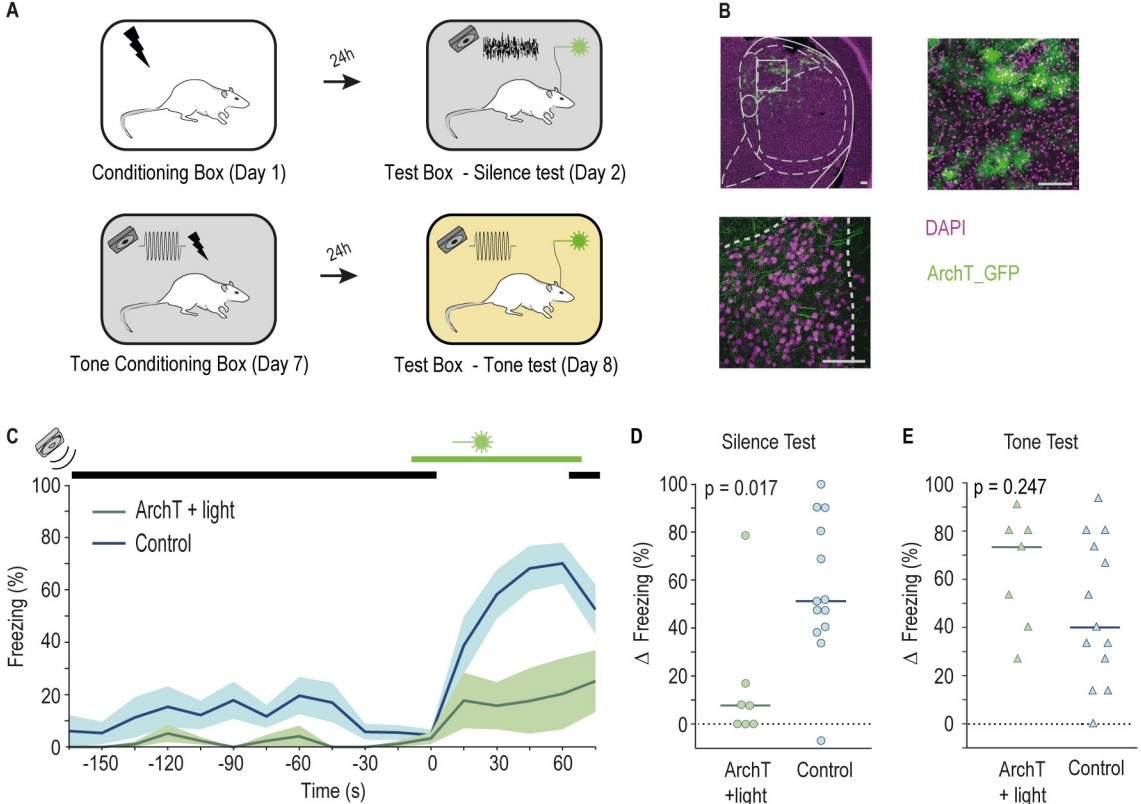

**Fig 3. Activity in the MGD is necessary for freezing triggered by the cessation of movement-evoked sound but not by a conditioned tone cue.** (A) Schematics of the behavioral paradigm used for testing the role of the MGD in freezing triggered by the cessation of movement-evoked sound and 5-kHz pure tone. (B) Top left, coronal brain section from a representative rat, in which MGD neurons express ArchT; top right, image with a higher magnification of the area highlighted by the white square on the left image (see S1 Movie for detailed anatomy). Bottom, axon terminals in the tip of LA originating from ArchT-expressing cells in MGD. Pink, cell bodies labeled with DAPI. Green, cell bodies and cellular processes of cells expressing ArchT-GFP. Scale bar 100 μm. (C and D) Same as Fig 1C and 1D (respectively) for rats expressing ArchT and exposed to light illumination in the MGD (green, $n = 7$) and respective controls (blue, $n = 13$ including animals from both Ct light and Ct ArchT groups). Horizontal bar represents the median value (ArchT + light = 7.73%, Control = 51.20%, Wilcoxon rank sum test, rank sum = 43) (S1 Data, Sheet Fig 3). (E) Individual value plot displaying the change in the percentage of time animals spent freezing during the 15 seconds immediately preceding the presentation of the 5-kHz conditioned tone and the 15 seconds during which the tone was played. Horizontal bar represents the median value (ArchT + light = 73.33%; Control = 40.00%; Wilcoxon rank sum test, rank sum = 88.5) (S1 Data, Sheet Fig 3). ArchT, archaerhodopsin from *Halorubrum* strain TP009; Ct ArchT, control ArchT; Ct light, control light; LA, lateral amygdala; MGB, medial geniculate body; MGD, dorsal division of the MGB; DAPI, 4′, 6-diamidino-2-phenylindole; GFP, green fluorescent protein.

optogenetically inhibiting the MGD (Fig 3A). Inactivation of the MGD did not affect the expression of freezing triggered by the conditioned pure tone (Fig 3E). To better compare the role of the MGD in freezing triggered by a conditioned tone or the cessation of movement-evoked sound, we analyzed the increase in freezing during the initial 15 seconds of the stimulus (silence gap or tone) in both tests. With this analysis, we diminish potential confounds related with differences in the duration of the stimulus and MGD inactivation (60-second silence and 75-second inactivation versus 15-second tone and 30-second inactivation). We found that after 15 seconds of exposure to either the silence gap or the conditioned stimulus, control animals show a significant increase in freezing (S4 Fig). In contrast, when MGD is inactivated (ArchT + light group), rats increased freezing upon the 15-second conditioned tone but not to the first 15 seconds of the silence gap (S4 Fig). Moreover, the same animals (ArchT + light) increase their freezing significantly more during the tone test than during the silence test, despite optogenetic inactivation of the MGD in both cases (S4 Fig). These results

show that activity in the MGD is necessary for the display of freezing triggered by the transition from movement-evoked sound to silence but not that triggered by a conditioned pure tone. In line with physiological data, this indicates that the MGD may be preferentially recruited to process the offset of sound [17,18,20]. Whether it is the offset cells in MGD that directly or indirectly drive activity in LA, leading to the expression of freezing, remains to be established. Furthermore, even though our manipulation affected mostly neurons in MGD, we cannot exclude the possibility that we have silence some neurons in suprageniculate nucleus (SG) and the dorsal region of the MGV (S3 Fig).

The MGD sends direct projections to the LA [15] but also projects to discrete areas in the cortex that through polysynaptic connections can convey information to the LA. To test whether auditory processing areas downstream of the MGD are necessary for the response to the cessation of the movement-evoked sound, we focused on the 2 regions in the temporal cortex to which MGD axons project: posterodorsal (PD) and ventral area (VA) [21,25]. We inactivated PD or VA using a pharmacological approach (S5 Fig). Briefly, rats with bilateral cannulas targeting PD or VA received a microinjection of the GABA agonist muscimol, just prior to the silence test (see Methods). At the end of the experiment, animals received an injection of the retrograde tracer cholera toxin B (CTB), which allowed confirmation that the inactivated cortical region received input from MGD (Fig 4C and 4F). Inactivation of PD with muscimol did not significantly decrease freezing triggered by silence in comparison with control rats (Fig 4A and 4B). Inactivating VA, however, strongly impaired freezing during the silence period (Fig 4D and 4E). These results show that activity in VA, but not PD, is necessary for freezing triggered by the cessation of movement-evoked sound.

## Discussion

Together, our results suggest that the MGD, VA, and LA form a network important for processing the transition from movement-generated sound to silence. Freezing triggered by the cessation of movement sound is likely to rely on cells with sound offset responses, although further experiments are necessary to confirm this hypothesis. Reports demonstrating the presence of sound offset neurons in the MGD and their output regions in the auditory cortex [17–20,26–28] support our proposed network. Indeed, a growing body of work has identified neurons with sound offset responses throughout the auditory system. Particularly relevant to this study is the presence of offset cells in the marginal zone of the MGV, including its dorsal region, which might have been affected by our inactivation experiments [18]. These reports suggest the existence of an "offset pathway" thought to be of major importance in processing speech, sound localization, and movement detection [29]. Importantly, in most of these studies, the offset responses are transient, lasting a fraction of a second, whereas in our study, freezing lasts for several seconds. One possibility is that cells with offset responses act as a trigger and downstream targets drive the long-lasting defensive responses. Alternatively, prior exposure to shock, a necessary condition for rats to freeze upon silence onset, may lead to plastic changes that mediate a change from transient to sustained sound offset responses in any of the identified brain regions. Although we identified 2 input stations to the LA, there are multiple scenarios by which information may flow in this circuit. One scenario is that of a feed-forward circuit whereby MGD drives VA, which, in turn, activates LA. This activation is probably indirect because evidence for direct projections from the VA to the amygdala is lacking [30] (but see [16]). Alternatively, feedback loops within the identified circuit may contribute to silence-triggered freezing. VA neurons project back to the MGD, allowing for top-down enhancement of relevant stimuli, possibly instructed by the amygdala. Indeed, neurons in LA that project back to primary auditory cortex are required for the expression of freezing to conditioned pure tones [31]. A similar feedback

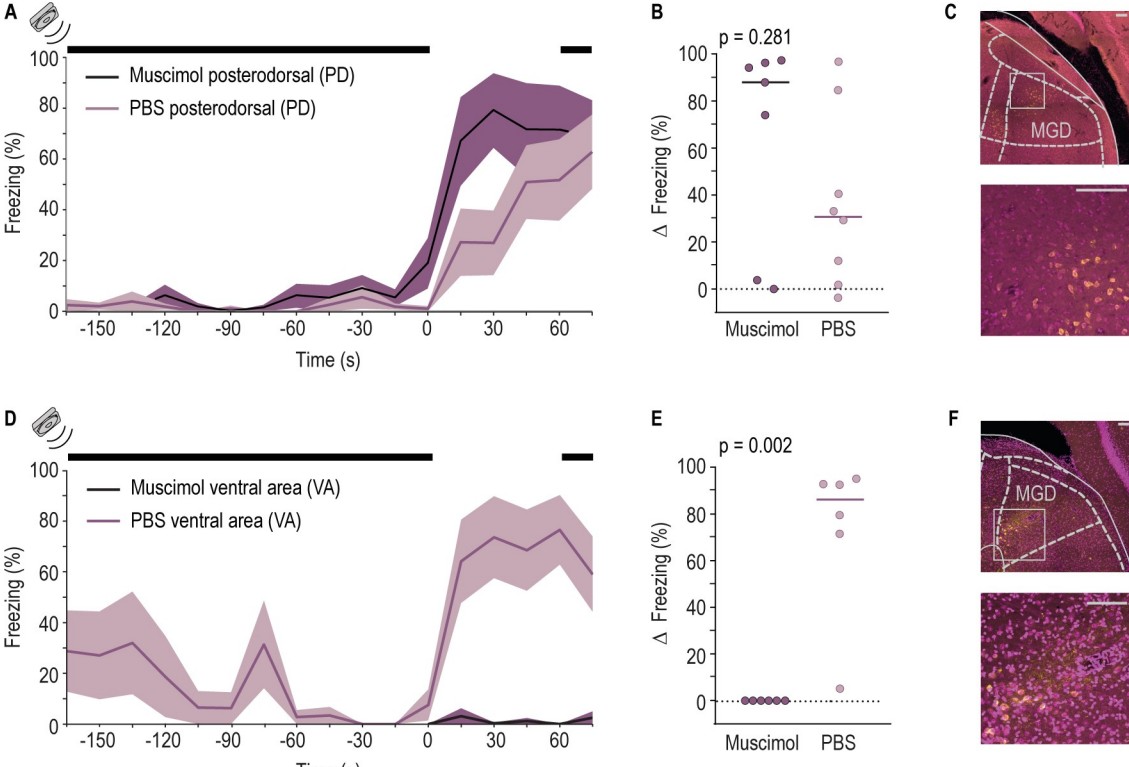

**Fig 4. The VA but not the PD region of the auditory cortex is necessary for freezing triggered by silence.** (A, B) Same as Fig 1C and 1D, (respectively) but for rats with bilateral injection of muscimol (dark purple, *n* = 7) or PBS (light purple, *n* = 8) in the area PD. Horizontal bar represents the median value of the group (muscimol = 87.87%, PBS = 30.73%, Wilcoxon rank sum test, rank sum = 54). (S1 Data, Sheet Fig 4). (C) Top, coronal brain section from a representative rat in which MGD cells are labeled with CTB injected in the PD. Bottom, image with higher magnification of the area highlighted by the white square on the top image. Pink, cell bodies labeled with DAPI. Yellow, cell bodies of CTB-positive cells. (D, E) Same as (A, B) (respectively), but for rats with bilateral injections of muscimol (dark purple, *n* = 6) or PBS (light purple, *n* = 6) in VA. Horizontal bar represents the median value (muscimol = 0.00%, PBS = 86.07%, Wilcoxon rank sum test, rank sum = 57). (F) Same as (C) but for CTB injections in the VA. Scale bar 100 μm. (S1 Data, Sheet Fig 4). ArchT, archaerhodopsin from *Halorubrum* strain TP009; Ct ArchT, control ArchT; Ct light, control light; CTB, cholera toxin subunit B; DAPI, 4', 6-diamidino-2-phenylindole; LA, lateral amygdala; MGB, medial geniculate body; MGD, dorsal division of the MGD; PBS, phosphate buffered saline; PD, posterodorsal; VA, ventral area.

loop has been proposed for the encoding of behavioral relevance or context of visual stimuli, whereby the LA would contribute to the formation of neuronal ensembles in visual cortex that encode stimulus relevance, independent of ensembles that encode visual stimulus identity [32].

The current study brings new insights into the mechanisms by which prey animals use the movement of others to infer danger, advancing our knowledge regarding how defense circuits respond to natural sound cues. By describing a pathway that includes auditory regions that have not been previously implicated in defense responses, it expands the network that processes auditory cues in the context of danger. More broadly, this study sets the stage to further our understanding of how sensory stimuli and their behavioral relevance are encoded by sensory systems and brain regions involved in motivated survival behaviors.

## Methods

### Ethical statement

The Champalimaud Center for the Unknown follows the National Law (Decreto-Lei 113/ 2013), European Directive 2010/63, and Federation of European Laboratory Animal Science

Associations (FELASA) guidelines and recommendations concerning laboratory animal experimentation and welfare. The experiments in this study were approved under the protocol ID 014304.

## Animal care

Naive male Sprague Dawley rats (300–350 g) for *c-fos* experiments and 200–250 g for optogenetic and muscimol experiments were obtained from a commercial supplier (Harlan and Charles Rivers, respectively). After arrival, animals were pair housed in Plexiglas top filtered cages and maintained on a 12-hour light/dark cycle with ad libitum access to food and water. All behavioral procedures were performed during the light phase of the cycle.

For the *c-fos* experiments, animals were kept in pairs and acclimated for at least 1 week before experimental manipulation. For optogenetic experiments, animals were separated 4–6 days after arrival and subjected to virus injection and/or optic fiber implantation surgery. After this procedure, animals were kept alone in Plexiglas boxes before experimental manipulation. For muscimol experiments, animals were separated on the fourth day after arrival and subjected to cannula implantation surgery. After the procedure, animals were kept alone for 7 days before experimental manipulations.

## Viral vector, neuronal tracer, and muscimol

The viral vector Adeno-associated virus expressing ArchT (AAV2/5 CAG-ArchT-GFP $1.3 \times x10^{12}$ vg/mL) was produced by and purchased to University of North Carolina (UNC) vector core facility. The viral vector was diluted with sterile phosphate buffered saline (PBS) immediately before injection to obtain a concentration of $4 \times 10^{11}$ vg/mL.

The transynaptic neuronal tracer Cholera Toxin subunit B Alexa Fluor 555 Conjugate (CTB Alexa 555, 1 mg/mL) was produced by and purchased from Life Technologies. The GABA agonist muscimol (Cat# 2763-96-4) was bought from Sigma-Aldrich.

## Stereotactic surgery

Animals were anaesthetized with isoflurane (Vetflurane 1,000 mg/g, Virbac) and placed in a stereotaxic apparatus (David Kopf Instruments). Small craniotomies were made using standard aseptic techniques.

For the LA inhibition experiments, animals assigned to the ArchT + light and Ct ArchT groups received injections targeted bilaterally to the LA (stereotaxic coordinates from Bregma, anterior–posterior: −3.3 mm, dorsal–ventral: −8.1, medial–lateral: 5.2 mm; derived from Paxinos and Watson 2007 [33]) using stainless steel guide cannula (24 gauge; Plastics One). Following cannula guide placement, 0.3–0.4 μL injections of rAAV2/5-CAG-ArchT-GFP were made through a stainless steel injection cannula (31 gauge; Plastics One), attached to a Hamilton syringe via polyethylene tubing. Injections were controlled through an automatic pump (PHD 2000; Harvard Apparatus). For animals assigned to the ArchT + light group optical fibers (200-μm, 0.37 numerical aperture, Doric Lenses) were implanted in the LA (stereotaxic coordinates from Bregma, anterior–posterior: −3.3 mm, dorsal–ventral: −8.15 or 8.05 mm, medial–lateral: 5.2 mm [33]) and affixed to the skull using stainless steel mounting screws (Plastics One) and dental cement (TAB 2000, Kerr). No fibers were implanted in animals assigned to the Ct ArchT group. Rats were kept for 4 weeks before any behavioral manipulation to allow maximal expression of the virus. Animals of the Ct light group were subjected to the same procedure, including implanting optical fibers, but no virus was injected.

For the dorsal medial geniculate nucleus (MGD) inhibition experiments, the procedures were similar, but injections were targeted bilaterally to the MGD (stereotaxic coordinates from

Bregma, anterior–posterior: −5.8 mm, dorsal–ventral: −5.3, medial–lateral: ±3.4 mm [33]) and 0.2 μL of rAAV2/5-CAG-ArchT-GFP was injected.

For the VA and PD inhibition experiments, animals were subjected to a stereotaxic surgery to bilaterally implant stainless steel guide cannulas (24 gauge, Plastics One) in either the PD (stereotaxic coordinates from Bregma, anterior–posterior: −5.7 mm, dorsal–ventral: −4.3 mm, and medial–lateral: +6.5 mm [33]) or VA (stereotaxic coordinates from Bregma, anterior–posterior: −3.5 mm, dorsal–ventral: −6.5 mm, medial–lateral: +6.5 mm [33]) areas of the auditory cortex. Following cannula implantation, a dummy cannula (31 gauge, Plastics One) was kept until test day.

For the retrograde tracing of auditory thalamus projections to these regions (used to confirm that the area that was inactivated indeed received input from the MGD), we took advantage of the already implanted guide cannulas, and posterior to the test day, CTB Alexa 555 (0.2 μL) was bilaterally injected in half of the experimental animals following the same protocol as the one used to inject the viral vector. Animals were humanely killed days after this procedure to allow sufficient transport of the tracer.

All injection sites and fiber placements were verified histologically, and rats were excluded if either were not on the correct place.

## Behavioral apparatus

Four distinct environments were used in this study: conditioning box, test box–silence test, tone conditioning box, and test box–tone test.

Both conditioning boxes (model H10-11R-TC, Coulbourn Instruments) have a shock floor of metal bars (model H10-11R-TC-SF, Coulbourn Instruments) and are placed inside a sound isolation chamber (Action Automation and Controls). In these boxes, a precision programmable shocker (model H13-16, Coulbourn Instruments) delivered the footshocks, and tones were produced by a sound generator (RM1, Tucker-Davis Technologies) and delivered through a horn tweeter (model TL16H8OHM, VISATON).

The test box–silence test consisted of a 2-partition chamber made of clear Plexiglas walls (Gravoplot) divided in 2 equal halves, placed inside a sound attenuation chamber. The test box–tone test consisted of a plastic box with a round platform in the bottom. Both test boxes had bedding on the floor. Tones were generated, delivered, and calibrated using the same apparatus as the conditioning boxes. To minimize generalization between environments, conditioning with shock (day 1) took place with light, and silence test (day 2) took place in the dark (Fig 1A). For tone conditioning experiments, tone conditioning took place in the dark, and recall took place under dim yellow light. Rats' behavior was tracked by a video camera mounted on the ceiling of the attenuating cubicle. A surveillance video acquisition system was used to record and store all videos on hard disk.

## Behavioral procedures

All rats were preexposed to the different boxes prior to the beginning of the experiments. To test the role of auditory motion cues, the previously recorded movement-evoked sound [4] was played during exposure to the test box–silence test and during the silence test. To generate this sound, 1 single rat was recorded as it moved around in 1 of the partitions of the social interaction box with bedding on the floor. The sound of this rat moving around was recorded through a microphone placed over the chamber (Avisoft-UltraSoundGate system 416H, microphone model CM16/CMPA). This allowed recording of ultrasounds. Sections of the recording without vocalizations were chosen for the playback.

## Optogenetic experiments

For the silence test experiments, rats were trained and tested as previously described [4]. During the test session, a fiber-optic cable terminating in 2 ferrules (Branching Fiber-optic 200 μm, 0.22 numerical aperture, Doric lenses) was connected to the chronically implanted optic fibers. Animals in the ArchT + light and Ct light groups received laser illumination (estimated 30 mW at the tip of the fiber) that started 10 seconds before the silence. Illumination lasted until 5 seconds after the resumption of the playback of the movement-evoked sound. After the test session, animals returned to their home cages.

To test the role of the MGD in recall after tone conditioning, the same animals that were tested to the silence were exposed to the tone conditioning box and test box–tone test during the following 4 days of experiment. On the fifth day, after a 5-minute baseline, animals received 3 tone–shock pairings (5-kHz tone, 70 dB, 15 seconds coterminating with 1-mA shock, 0.5 seconds). Animals were tested to the sound the day after in the test box–tone test, in which animals were exposed to 3-tone presentation, (5-kHz tone, 70 dB, 15 seconds). Laser illumination was identical to the one used in LA experiments.

## c-fos experiments

Experiments followed a modified protocol of the silence test experiments (described previously), in which the test session lasted for 9 minutes. In these experiments, animals in the silence group were exposed to 2 periods of silence with the duration of 1 minute each (and with a period of 3 minutes separation), whereas for the control group (continuous sound), the recorded sound of movement was played throughout the entire test session.

## Muscimol experiments

Experiments followed the same protocol as the silence test experiments; however, in the test day, animals were injected with 0.2 μl muscimol (0.5 μg/μl concentration at 0.1 μl/min rate) in either the VA or PD. The injections were performed with the help of an injection cannula, attached to a Hamilton syringe controlled by an automatic pump (PHD 2000; Harvard Apparatus). Animals were left undisturbed for a period of 1 hour and 30 minutes after which they were tested.

## Histology

Animals were deeply anesthetized with pentobarbital (600 mg/kg, i.p.) and transcardially perfused with PBS (0.01 M), followed by ice-cold 4% paraformaldehyde (Paraformaldehyde Granular; cat#19210; Electron Microscopy Sciences) in 0.1 M phosphate buffer (PB) (PFA-PB). Brains were removed and postfixed in 4% PFA-PB and kept at 4°C.

Coronal sections containing the LA (2.50 mm to 3.80 mm posterior to Bregma), VA (3.20 mm to 4.60 mm posterior to Bregma), PD (5.40 mm to 6.7 mm posterior to Bregma), and/or the MGB of the auditory thalamus (5.60 mm to 6.50 mm posterior to Bregma) were cut and mounted.

Mounted slices were rehydrated for 10 minutes with PBS and 500 μl of a 1:1,000 diluted 4′, 6-diamidino-2-phenylindole (DAPI) solution (D9542, Sigma-Aldrich) was applied to each slide and incubated for 20 minutes in a shaker (10 rpm). Slices were then washed 3 times with PBS and rinsed with distilled water. Slices were coverslipped with Moviol 4–88 (81381, Sigma-Aldrich).

Fluorescent images were taken with the confocal microscope Zeiss LSM 710, Axioimager2, Objective Plan-Apochromat 20x/0.8M27, and software Zen 2010. Pictures were processed in

ImageJ for compiling z-stacks, and exposure adjustments (same adjustment for both channels) were made with Photoshop.

Immunohistochemistry experiments examining *c-fos* expression in the MGB were performed in brain slices from animals that were rapidly and deeply anaesthetized with pentobarbital (600 mg/kg, i.p.) 2 hours after the beginning of the behavioral paradigm. While anesthetized, animals were transcardially perfused with PBS, followed by ice-cold 4% PFA-PB. Brains were removed and postfixed in 4% PFA-PB for 24 hours and subsequently cryoprotected in 20% glycerol (J.T.Baker) in 0.1 M PB for 72 hours at 4˚C. Using a sliding microtome, sections of 40 μm containing the auditory thalamus (5.40 to 6.40 posterior to Bregma) were cut and collected in PBS. Next, sections were transferred to a 0.1% sodium azide (Sigma-Aldrich) in PBS solution for storage. The immunohistochemical staining was performed simultaneously for all brain sections analyzed. Staining was performed in free-floating sections. Sections were washed 3× for 10 minutes with PBS, incubated for 10 min with 0.9% $H_2O_2$ (Sigma-Aldrich), washed again 3×10 minutes in PBS and blocked in PBS with 1% bovine serum albumin (BSA) (cat#A7906, Sigma-Aldrich) and 0.1% Triton X-100 for 1 hour at RT. Slices were then incubated O.N. RT with anti c-fos 1ry AB (1:500; rabbit; sc-52 Santa Cruz Biotechnology) in PBS with 1% BSA and 0.1% Triton X-100. The next morning, sections were washed 3× for 10 minutes with PBS and incubated with goat anti-rabbit biotinylated 2ry antibody (1:1,000; Cat#405008, Southern Biotec) in PBS with 1% BSA and 0.1% Triton X-100 for 1 hour at RT. Sections were washed 3× for 10 minutes in PBS, incubated with horseradish peroxidase streptavidin (1:300; cat#SA-5004; Vector Laboratories) in PBS with 0.2% Triton X-100 for 1 hour RT, washed 3× for 10 minutes in PBS-B, and developed in diaminobenzidine tablets (DAB) (cat# D4418-50 SET; Sigma) for 3 minutes.

Sections were mounted on electrostatic slides, air dried, dehydrated in ethanol and xylenes, and coverslipped with DPX. Bright-field images were taken in Zeiss Axioimager M1 microscope equipped with a CCD camera (Hamamatsu C8484), with objective 20×/0.80. Sections from comparable anterior-posterior levels were selected for scoring, and images from each subnucleus were taken systematically from the same area based on distance from reference points specific for each anterior-posterior level. Cell counts were scored using NIH Image J. For the initial analysis, cell counts for each subnucleus of the thalamus were averaged into a single score for each rat. For the analysis along the anterior-posterior axis of the MGD, we divided the sections in anterior (sections including and posterior to 5.64 until 5.76 [including] relative to Bregma), medial (sections posterior to 5.76 until 6.00 [including] posterior to Bregma), and posterior (sections posterior to 6.00 until 6.24 [including] relative to Bregma) and averaged the *c-fos*—labeling cells in those slices.

## Analysis of behavioral data

Animals' freezing behavior during the silence test was automatically scored using FreezeScan from Clever Sys. Baseline freezing levels were calculated using the median percentage of freezing during the 60 seconds preceding the onset of silence. To avoid confounding factors, such as freezing, triggered by other cues that were not the onset of silence, animals were considered outliers and thus excluded if their baseline freezing was higher than the third quartile + 1.5 (third quartile–first quartile) baseline freezing of the population used in each experiment (LA, MGD, VA, and PD experiments were considered separately). Because of the noise generated by the shutter used in the optogenetics experiments, animals that were freezing more than 50% in the 10 seconds period between the opening of the shutter and silence onset were also excluded.

## Statistical analysis

For data analysis of the silence test behavioral experiments, we focused on the percentage of freezing during the silence gaps (that lasted 1 minute) and used as baseline the minute immediately preceding the silence interval. For data analysis of the tone test behavioral experiments, we analyzed data in a similar way but focused on the period of tone presentation (that lasted 15 seconds) and used as baseline the 15 seconds immediately preceding it. In this manner, we ensure that both measures have the same sampling time.

A Shapiro–Wilk test was used to access the normality of our data. The behavioral data were, in general, not normally distributed, and sample sizes were small, so we used nonparametric tests only.

For comparisons between groups (when comparing the change in freezing between baseline and silence periods), we used a Wilcoxon–Mann–Whitney test. For comparisons within group (comparing percentages of freezing during baseline and silence), we conducted a Wilcoxon signed-ranked test. For the *c-fos* behavioral data, we averaged the percentage of freezing during the 2 periods of 1 minute preceding the silence inserts and the 2 periods of silence. For comparisons within the group (silence or continuous sound), we conducted a Wilcoxon signed-ranked test. For comparisons between groups of the *c-fos*-labeled cells, our data were not normally distributed, and therefore, we used a Mann–Whitney test. For analyzing the effect of position along the anterior-posterior axis and exposure to silence in the expression of *c-fos*, we conducted a 2-way ANOVA for unbalanced design given that our data were normally distributed. Statistical analysis was performed using MATLab.

## Supporting information

**S1 Fig. The LA is necessary for the expression of freezing driven by the cessation of movement-evoked sound.** (A) Coronal slices representing fiber placements and spread of virus expression for ArchT + light group. (B) Fiber placement and/or injection site for the Ct light (light blue) and Ct ArchT (dark blue) groups. (C) Same as Fig 1D, but for animals of the 2 control groups. Horizontal bar represents the median value of the group (Ct light = 66.13%, Ct ArchT = 73.67%, Wilcoxon rank sum test, rank sum = 72) (S1 Data, Sheet S1 Fig). (D–F) Line graph showing average time spent freezing during the minute immediately preceding the cessation of the movement-evoked sound (baseline) and the minute of silence for each rat of the ArchT + light ($n = 7$), Ct light ($n = 9$), and Ct ArchT ($n = 6$) groups. Wilcoxon signed rank test. Baseline versus silence ArchT + light signedrank = 1; Ct light signedrank = 1; Ct ArchT signedrank = 0. (S1 Data, Sheet S1 Fig). ArchT, archaerhodopsin from *Halorubrum* strain TP009; Ct light, control light; Ct ArchT, control ArchT; LA, lateral amygdala.
(TIF)

**S2 Fig. Representative pictures of c-*fos*-labeled cells in MGB.** Representative images of coronal slices from rats exposed to silence gaps showing *c-fos* labeling in different subnuclei of the MGB. In particular, we observe *c-fos*-positive cells in the shell of the MGV, an area particularly responsive to the offset of sounds. *c-fos*-positive cells in this region were not quantified because it does not project directly to the LA. LA, lateral amygdala; MGB, medial geniculate body; MGV, ventral division of the MGB.
(TIF)

**S3 Fig. The MGD is necessary for the expression of freezing driven by the cessation of the movement-evoked sound but not by a discrete auditory cue.** (A) Coronal slices representing fiber placements and spread of virus expression for ArchT + light group. (B) Fiber placement and/or injection site for the Ct light (light blue) and Ct ArchT (dark blue) groups. (C)

Representative image of cells in MGD expressing ArchT-GFP. (D) Same as Fig 3D but for animals of the 2 control groups. Horizontal bar represents the median value (Ct light = 49.47%, Ct ArchT = 68.93%, Wilcoxon rank sum test, rank sum = 48) (S1 Data, Sheet S3 Fig). (E–G) Line graph showing average time spent freezing during the minute immediately preceding the cessation of the movement-evoked sound (baseline) and the minute of silence for each rat of the ArchT + light ($n = 7$), Ct light ($n = 8$), and Ct ArchT ($n = 5$) groups with surgeries targeting the MGD. Wilcoxon signed rank test, baseline versus silence ArchT + light signedrank = 0; Ct light signedrank = 1; Ct ArchT signedrank = 0. (S1 Data, Sheet S3 Fig). ArchT, archaerhodopsin from *Halorubrum* strain TP009; Ct light, control light; Ct ArchT, control Arch T; GFP, green fluorescent protein; LA, lateral amygdala; MGB, medial geniculate body; MGD, dorsal division of the MGB.
(TIF)

**S4 Fig. Effect of MGD inactivation during the first 15 seconds of the silence test in comparison with tone test.** (A) Line graph showing average time spent freezing during the 15 seconds immediately preceding the cessation of the movement-evoked sound or tone (baseline) and the 15 seconds of stimulus (silence or tone) for each rat of the ArchT + light ($n = 7$) and Control ($n = 13$) groups. Wilcoxon signed rank test, baseline versus silence ArchT + light signedrank = 0; Control signedrank = 1; baseline versus tone ArchT + light signedrank = 0; Control signedrank = 0 (S1 Data, Sheet S4 Fig). (B) Individual dots represent the change in the percentage of time each animal spends freezing during the 15 seconds immediately preceding the cessation of the movement-evoked sound or tone (baseline) and the 15 seconds of stimulus (silence or tone) for each rat of the ArchT + light ($n = 7$) and Control ($n = 13$) groups. Horizontal bar represents the median value of the group (Silence test ArchT + light = 0.00%, Control = 37.07%; Tone test ArchT + light = 73.33%, Control = 40%). Kruskal–Wallis test chi-squared = 8 (S1 Data, Sheet S4 Fig) (S1 Data, Sheet S4 Fig). ArchT, archaerhodopsin from *Halorubrum* strain TP009; MGB, medial geniculate body; MGD, dorsal division of the MGB.
(TIF)

**S5 Fig. Coronal slices representing injection site** for PBS (light purple) and muscimol (dark purple) in (A) PD and (B) VA. Areas shaded in gray correspond to PD and VA (respectively). PBS, phosphate buffered saline; PD, posterodorsal; VA, ventral area.
(TIF)

**S1 Data. Raw data file.**
(XLSX)

**S1 Movie. Neurons in MGD expressing ArchT-GFP.** ArchT, archaerhodopsin from *Halorubrum* strain TP009; GFP, green fluorescent protein; MGD, dorsal division of the medial geniculate body.
(MP4)

## Acknowledgments

We thank Alexandra Medeiros and Andreia Cruz for their input to the project and Alexandra in particular for her effort in the histological characterization of the MGD. We also want to thank all the members of the Behavioral Neuroscience lab, as well as Samuel Walker and Tiago Marques, for comments on the manuscript. We thank Glenn Schafe and Stephanie Maddox for the training in immunohistochemical techniques to detect the expression of immediate early genes. This work was developed with the support from the Champalimaud Histopathology platform and the research infrastructure Congento LISBOA-01-0145-FEDER-022170.

## Author Contributions

**Conceptualization:** Ana G. Pereira, Marta A. Moita.

**Formal analysis:** Ana G. Pereira, Matheus Farias, Marta A. Moita.

**Funding acquisition:** Marta A. Moita.

**Investigation:** Ana G. Pereira, Matheus Farias.

**Methodology:** Ana G. Pereira, Marta A. Moita.

**Project administration:** Marta A. Moita.

**Resources:** Marta A. Moita.

**Supervision:** Marta A. Moita.

**Visualization:** Ana G. Pereira, Marta A. Moita.

**Writing – original draft:** Ana G. Pereira.

**Writing – review & editing:** Ana G. Pereira, Marta A. Moita.

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
