## [Editor Report · Decision Letter 0]

9 Sep 2019

Dear Dr Moita, 

Thank you for submitting your manuscript entitled "A newly identified auditory circuit is involved in processing a natural sound cue of danger" for consideration as a Short Report by PLOS Biology.

Your manuscript has now been evaluated by the PLOS Biology editorial staff, as well as by an Academic Editor with relevant expertise, and I am writing to let you know that we would like to send your submission out for external peer review.

Please re-submit your manuscript within two working days, i.e. by Sep 11 2019 11:59PM.

Kind regards,

Gabriel Gasque, Ph.D.,

Senior Editor

PLOS Biology

---

## [Decision Letter · Decision Letter 1]

14 Oct 2019

Dear Dr Moita,

Thank you very much for submitting your manuscript "A newly identified auditory circuit is involved in processing a natural sound cue of danger" for consideration as a Short Report at PLOS Biology. Your manuscript has been evaluated by the PLOS Biology editors, by an Academic Editor with relevant expertise, and by three independent reviewers. You will note that reviewer 3, Jennifer Linden, has signed her comments. 

In light of the reviews (below), we will not be able to accept the current version of the manuscript, but we would welcome resubmission of a much-revised version that takes into account the reviewers' comments. We cannot make any decision about publication until we have seen the revised manuscript and your response to the reviewers' comments. Your revised manuscript is also likely to be sent for further evaluation by the reviewers.

Your revisions should address the specific points made by each reviewer. As you will see, reviewer 3 could be potentially satisfied with additional analyses and clarifications. However, reviewers 1 and 2 think that additional experiments are needed. Having discussed these comments with the Academic Editor, we think that for your revision to be successful at PLOS Biology, you need to include the additional data requested, but paying special attention to the concerns raised by reviewers 1 and 3.

Please note also note that Short Reports should be four main figures or less. Therefore, please either add the new data to existing figures or to supporting information. 

Please submit a file detailing your responses to the editorial requests and a point-by-point response to all of the reviewers' comments that indicates the changes you have made to the manuscript. In addition to a clean copy of the manuscript, please upload a 'track-changes' version of your manuscript that specifies the edits made. This should be uploaded as a "Related" file type. You should also cite any additional relevant literature that has been published since the original submission and mention any additional citations in your response. 

Before you revise your manuscript, please review the following PLOS policy and formatting requirements checklist PDF: http://journals.plos.org/plosbiology/s/file?id=9411/plos-biology-formatting-checklist.pdf. It is helpful if you format your revision according to our requirements - should your paper subsequently be accepted, this will save time at the acceptance stage.

Please note that as a condition of publication PLOS' data policy (http://journals.plos.org/plosbiology/s/data-availability) requires that you make available all data used to draw the conclusions arrived at in your manuscript. If you have not already done so, you must include any data used in your manuscript either in appropriate repositories, within the body of the manuscript, or as supporting information (N.B. this includes any numerical values that were used to generate graphs, histograms etc.). For an example see here: http://www.plosbiology.org/article/info%3Adoi%2F10.1371%2Fjournal.pbio.1001908#s5.

For manuscripts submitted on or after 1st July 2019, we require the original, uncropped and minimally adjusted images supporting all blot and gel results reported in an article's figures or Supporting Information files. We will require these files before a manuscript can be accepted so please prepare them now, if you have not already uploaded them. Please carefully read our guidelines for how to prepare and upload this data: https://journals.plos.org/plosbiology/s/figures#loc-blot-and-gel-reporting-requirements.

Upon resubmission, the editors will assess your revision and if the editors and Academic Editor feel that the revised manuscript remains appropriate for the journal, we will send the manuscript for re-review. We aim to consult the same Academic Editor and reviewers for revised manuscripts but may consult others if needed.

We expect to receive your revised manuscript within two months. Please email us (plosbiology@plos.org) to discuss this if you have any questions or concerns, or would like to request an extension. At this stage, your manuscript remains formally under active consideration at our journal; please notify us by email if you do not wish to submit a revision and instead wish to pursue publication elsewhere, so that we may end consideration of the manuscript at PLOS Biology.

When you are ready to submit a revised version of your manuscript, please go to https://www.editorialmanager.com/pbiology/ and log in as an Author. Click the link labelled 'Submissions Needing Revision' where you will find your submission record. 

Sincerely,

Gabriel Gasque, Ph.D., 

Senior Editor

PLOS Biology

Reviewer remarks:

Reviewer #1: Pereira and colleagues describe a circuit of three nuclei involved in the freezing behaviour after hearing the cessation of sound made by conspecifics. They used a combination of pharmacological and optogenetic silencing together with behavioural experiments. C-fos and tracing data were used to corroborate the results. Overall, the manuscript is short, straight forward and interesting to a wide range of scientists. I have three major concerns that I would ask the authors to address and a number of minor comments that can be easily fixed listed below.

Major concerns:

1. The ArchT expression was not driven by a MGD specific cre-line. So the specificity of the optogenetic silencing was entirely based on the correct (and spatially restricted) injection of the viral vector into the MGD. As the MGD is small and very close to other potential contributors to this circuit like the other parts of the MGB, which can also generate offset responses (Anderson & Linden, 2016), the injection site and specificity needs to be validated. In fact, the results of the silencing look almost too good to be true, which made me wonder if the ArchT actually silenced larger parts of the MGB than just the MGD. This would still be a good, new and interesting result – but would not make claims that might not be justified. Please either show the injection sites of all tested animals to make sure ArchT is expressed only in MGD. Or change the wording to have the whole MGB be part of that circuit, rather than just MGD. Or repeat a couple of the experiments with a cre-dependent ArchT expression in a cre-line that is explicitly expressed in MGD. 

2. What is the spectrum/level of the movement evoked sound? Is it recorded from one or multiple rats? Are you sure there is no vocalization emitted in addition to plain movement sound – need to be tested in ultrasonic range. Please include this information in the methods.

3. How can you be sure that the increase in c-fos expression is really due to exposure to silent gaps? It could just be increased neural activity to any change in stimulus, while the ongoing sound is perceived as “boring” background. Has this been tested with another sound paradigm: e.g. ongoing noise as control (as you have done before), and then some other stimulus where you may have modulated noise instead of silence?

Minor comments:

There are groups which question the validity of the muscimol inactivation of the cortex too, but I would let this one go here.

Line 75: there is a question mark sign – should be an arrow (maybe just say: “White arrow shows…”)

Fig. 2C: Please provide higher magnification images so that the reader can see neurons rather than what looks like speckles of dirt. A counterstain for e.g. VGLUT or MAP2 would be also very helpful.

Line 166, 175, 359, 361: typo: kHz

Lines 173/177: ranksum or rank sum – be consistent!

Fig. 4: please indicate in the figure where the PBS or muscimol was injected (e.g. muscimol (into PD)), otherwise the reader needs to scramble though the legend first to get the message of the figure…

Why were only male mice used? Many journals now insist that both sexes were used, if not a scientific reason prevents the use on both.

Line 259, 349, 351: typo: fiber

Lines 373/374: use dots as decimals

Reviewer #2: Ana et al., reported that an auditory cue, the cessation of movement-evoked sound, induced defensive freezing, and further identified the underlying neural circuits comprising of MGd, ACx and LA. The authors thought this is a unique offset pathway for “cessation of movement-evoked sound” induced defense behavior. The study is mostly based on the behavioral analysis. However, there is major problem in their behavioral design. The authors’ interpretation of the behavior results and the conclusion could be completely wrong. The proposed new circuit is the classical circuit for fear conditioning.

1 The defense behavior induced by “cessation of movement-evoked sound” examined in this study was first reported by this research group in 2012 in current biology. This is not an innate behavior (as suggested for impending danger), but learned defense as it required conditioning training with foot shock (Fig. 1a). 

2 This is more problematic for the interpretation of the whole study. This training can directly establish an association between “cessation of movement-evoked sound” (e.g. caused by the trainee rat) and footshock, i.e. some fear conditioning. That means the nature of the later defense behavior could be simply a learned defense under a context-cue of “cessation of movement-evoked sound”. This possibility also explains a similar behavior (in a different context) previously reported by authors in 2012. Unfortunately, this was not examined for these years.

3 This alternative interpretation explains why the proposed neural circuits for this behavior is in fact the same as that for fear conditioning, including VA, LA, MGB etc. It is not a new circuit.

4 The authors should at least establish some correlation between neuronal activity in LA, VA, MGd, with the behavior. Is this really an offset responding pathway? How neurons in LA,VA, MGd respond during the cessation of movement-evoked sound and the following 1 minute silence. 

5 Experiments with specific silencing of projections from VA and MGD to LA would be helpful. Is LA activation sufficient for inducing freezing in those trainee rats?

Reviewer #3, Jennifer Linden: "A newly identified auditory circuit is involved in processing a natural sound cue of danger"

Pereira AG, Farias M and Moita MA

This article documents the discovery of an auditory circuit that drives freezing following cessation (offset) of movement-evoked rustling sounds, such as would be produced by the freezing of another animal. The circuit involves the dorsal subdivision of the medial geniculate body of the auditory thalamus (MGD), the ventral area of the auditory cortex (VA), and the lateral amygdala (LA). The authors use combined optogenetic and behavioural studies to demonstrate that this circuit is necessary to produce offset-evoked freezing behaviour.

This article is likely to be of wide scientific interest, because it is the first demonstration (to my knowledge) of a neural circuit driving behavioural responses to sound offset (rather than sound onset). Neural responses to sound offsets have been reported throughout the central auditory system, but the perceptual significance of these responses is not yet well understood. This paper provides clear evidence that higher central auditory brain areas contribute to behavioural responses to sound offsets. Moreover, this paper is extremely well-written: systematic and thorough yet succinct. Well done!!

I have a few major comments and several minor suggestions.

MAJOR COMMENTS:

(1) Lines 104-106: "Interestingly, prior electrophysiology studies in anaesthetised rodents showed a higher prevalence of sound offset responses in the MGD relative to other sub-nuclei of the auditory thalamus and higher number of offset cells in the caudal part of MGD." This is potentially misleading. It is true that previous studies have found evidence for offset responses in the MGD, but strong evidence for offset responses has also been reported in the MGV, especially in the "lateral shell" of the MGV. See for example:

He (2001) J Neurosci 21:8672–8679

He (2002) J Neurophysiol 88:2377–2386

Anderson and Linden (2016) J Neurosci 36:1977–1995

The authors should clarify that strong evidence for offset responses has also been reported in MGV, and also explain why this subdivision was not studied here (presumably because it does not project both directly and indirectly to the amygdala?).

(2) Figure 2b: Please comment in the text on the high variance in the MGD data. Is it possible that outliers in the "MGD" cases are at an extreme edge of this subdivision, on the border with other subregions, such as the MGV? See first major comment above.

(3) Figure 2 and associated text: It is quite confusing to use the term "silence" to mean an otherwise continuous sound interrupted by two silent gaps. The stimulus was definitely not "silence". Perhaps other terminology could be used here, e.g. "sound with silent gaps".

(4) Figure 3: Comparison between positive results in Fig. 3d ("silence test") and negative results in Fig. 3e ("tone test") is potentially flawed. The "silence test" results show the change in the percentage of time the animals spent freezing during the 1min immediately preceding the cessation of the movement sound and the 1min immediately following cessation. The tone test results show the change in percentage of time the animals spent freezing during the 15sec immediately before the tone and the 15sec of tone presentation. The tone test is likely to be underpowered relative to the silence test, because the time periods used for measuring freezing behaviour were much shorter in the tone test. To address this issue, results for the silence test could be re-computed using 15sec intervals, and either shown or at least mentioned in the text. Also: presumably the laser illumination *timing* must have been different for the silence test and the tone test, with a much longer period of optogenetic activation in the silence test than the tone test? This should also be mentioned as a possible reason for caution about interpretation of the negative result for the tone test. It is possible that the experiment and the analysis were simply less sensitive for the tone test, hence the negative result.

(5) In all of the figures shown, the Control data is pooled from Control-light and Control-ArchT conditions. While it is admirable that the authors tried two different control conditions, the Control-light condition is the far more critical one. Do all results hold when comparisons (e.g. of percent change in freezing) are made between ArchT-light and only Control-light conditions? Supplementary Figures seem to indicate that this is the case, but it would be worth stating in the text for every analysis shown in the main figures. It is especially important to mention whether results held for comparisons to the Control-light condition alone because of the point mentioned in the Methods section, lines 439-442: "Due to the noise generated by the shutter used in the optogenetics experiments, animals that were freezing more than 50% in the 10sec period between the opening of the shutter and silence onset were... excluded". In the tone test (Fig. 3e), the 10sec period between the opening of the shutter and tone cue onset would be 66% of the total pre-tone period (compared to 17% of the pre-silence period in the silence test); was the same exclusion criterion used for the tone test as for the silence test? 

MINOR CORRECTIONS:

line 64: "rats freezing behaviour" -- should be "rats' freezing behaviour"

line 73 (Fig. 1 legend): More information should be added here to aid interpretation of the mmunohistochemical images: e.g., "Pink, cell bodies labelled with DAPI; green, diffuse LA terminals labelled with ArchT."

line 75 (Fig. 1 legend): missing symbol for indicating tip of injector --- arrow?

line 77 (Fig. 1 legend): Please clarify here that Control condition includes both Control-light and Control-ArchT conditions. This is mentioned in the text but should also be mentioned in the legend to avoid confusion. See also major comment about mentioning results for Control-light comparison alone

line 147: figure reference should be to Figure 3c and 3d

line 148: figure reference should be to Figure 2c, 2d and 2e

lines 148-149: Anatomical and physiological properties of the MGD in rodents have also been addressed in:

He (2002) J Neurophysiol 88:2377-2386

Zhang, Yu, Liu, Chan and He (2008) Neuroscience 151:293-302

Anderson and Linden (2011) Hear Res 274:48-60

lines 205-208 and lines 210-211 (Fig. 4 legend): More information should be provided here to aid interpretation of the images, e.g. regarding subdivisions of amygdala indicated with dashed lines and localization of anterogradely labelled neurons from MGD or VA.

line 259 and elsewhere in methods: "optic fibber" should be "optic fiber"

line 262: "forth day" should be "fourth day"

line 271: "Phosphate Basal Solution" should be "Phosphate Buffered Saline", presumably??

line 275: "bought to Sigma-Aldrich" should be "bought from Sigma-Aldrich"

line 282: "groups were targeted bilaterally" should read "groups received injections targeted bilaterally"

line 312: "to this regions" should be "to these regions"

line 415: "byotinilated" should be "biotinylated"

---

## [Decision Letter · Decision Letter 2]

3 Feb 2020

Dear Dr Moita,

Thank you for submitting your revised Short Report entitled "Thalamic, cortical, and amydgala involvement in the processing of a natural sound cue of danger" for publication in PLOS Biology. I have now obtained advice from original reviewers 1 and 3, and have discussed their comments with the Academic Editor. You will note that reviewer 1, Conny Kopp-Scheinpflug, has signed her comments.

We're delighted to let you know that we're editorially satisfied with your manuscript. However before we can formally accept your paper and consider it "in press", we also need to ensure that your article conforms to our guidelines. A member of our team will be in touch shortly with a set of requests. As we can't proceed until these requirements are met, your swift response will help prevent delays to publication. Please also make sure to address the data and other policy-related requests noted at the end of this email.

*Copyediting*

*Published Peer Review History*

*Early Version*

*Submitting Your Revision*

Sincerely,

Gabriel Gasque, Ph.D., 

Senior Editor

PLOS Biology

ETHICS STATEMENT:

-- Please include the ID number of your protocol approved by the Champalimaud Center for the Unknown.

DATA POLICY:

--Please double check your entries in the S1 Data file, as some of these seem to be mislabeled; for example, in Fig 2, data for panel d is not included, in Fig 3, there are two entries for panel d but none for panel e.

--Please ensure that the figure legends in your manuscript include information on where the underlying data can be found and ensure your supplemental data file/s has a legend.

Reviewer remarks:

Reviewer #1, Conny Kopp-Scheinpflug: I would like to commend the authors to their revisions and a fine paper. They have properly addressed my initial concerns. This is an interesting follow-up of their 2012 study and will be of interest to a broad readership.

Reviewer #3: The authors have done a very good job with addressing the reviewers' comments, including with some useful additional supplementary figures. I have no additional queries.

---

## [Editor Report · Decision Letter 3]

9 Apr 2020

Dear Dr Moita,

On behalf of my colleagues and the Academic Editor, Jennifer K Bizley, I am pleased to inform you that we will be delighted to publish your Short Reports in PLOS Biology. 

Early Version

PRESS 

Kind regards,

Alice Musson

Publishing Editor, 

PLOS Biology

on behalf of

Gabriel Gasque,

Senior Editor

PLOS Biology